# A comparative analysis of COVID-19 physical distancing policies in South Africa and Uganda

**Sana Mohammad** [1]*, **Emma Apatu**[1], **Lydia Kapiriri** [2], **Elizabeth Alvarez** [1]

**1** Department of Health Research Methods, Evidence, and Impact, McMaster University, Hamilton, Ontario, Canada, **2** Department of Health, Aging and Society, McMaster University, Hamilton, Ontario, Canada

* mohams28@mcmaster.ca

## Abstract

COVID-19 responses internationally have depended on physical distancing policies to manage virus transmission, given the initial absence of treatments and limitations on vaccine availability. Different jurisdictions have different contexts affecting their responses such as past epidemic experience, ratings of epidemic preparedness, and income level. COVID-19 responses in African countries have not been well-studied. A qualitative multiple embedded case study design was used to examine the COVID-19 policies in South Africa and Uganda from January 2020 to November 2021. This study included a documentary review using government websites and reports, news articles, and peer-reviewed journal articles to obtain data on policy responses and contextual factors. Epidemiological data were collected from public sources. Key informant interviews with relevant stakeholders were used to confirm findings and cover missing information. A comparative analysis was conducted to explore differences in implementation of different types of physical distancing policies and potential consequences of lifting or prolonging public health measures. South African and Ugandan policy responses included physical distancing measures such as lockdown, international travel bans, school closures, public transportation measures, and curfew, in addition to socioeconomic relief programs and vaccinations. Differences between jurisdiction policy responses existed in terms of overarching strategy, timing, and stringency. This study provided in-depth comparisons of COVID-19 policy responses and relevant contextual factors in South Africa and Uganda. The study showed how contextual factors such as population age, geographic distribution, and recent epidemic response experience can influence COVID-19 transmission and response. The study also showed differences in overall strategy, timing, and strictness of epidemic management policies in these jurisdictions. These findings suggest it may be important to have sustained, strict measures to limit the spread of COVID-19 and manage the course of a pandemic, which need to be further explored alongside other important social and economic pandemic outcomes.

## Introduction

COVID-19 responses internationally have depended on physical distancing policies to mitigate or contain the spread of the virus since a pandemic was declared by the World Health

**Data Availability Statement:** Epidemiological data on COVID-19 cases, deaths, and vaccinations are available publicly and can be downloaded from: https://ourworldindata.org/coronavirus A policy-relevant database of contextual factors and policies

and in-depth individual country case reports used to inform this comparative analysis are shared publicly on the COVID-19 Policies and Epidemiology Working Group website: https://covid19-policies.healthsci.mcmaster.ca/ As part of the consent process, participants were informed that their interviews would remain confidential; therefore, original qualitative data are not available in any public repositories. Relevant excerpts from key informant interviews are included in the main text of the paper.

**Funding:** The authors received no specific funding for this work.

**Competing interests:** The authors have declared that no competing interests exist.

Organization (WHO) in March 2020. In the initial absence of treatments, and with limitations on vaccine availability and rollout, policy approaches have been the main strategy for governments managing the pandemic [1]. These policy approaches have centered on different types of physical distancing measures to separate people and limit the ability of the virus to spread [2].

Various factors have been associated with becoming infected or having severe outcomes from COVID-19 such as old age, gender, and comorbidities including smoking, obesity, occupational exposure, or being immunocompromised [3–6]. In addition to individual-level factors that may predispose individuals to COVID-19, it is possible that other contextual factors such as the type of government, trust in government, and availability of economic relief can influence compliance with public health measures designed to prevent individuals from becoming infected [7].

In light of governments' reliance on policy responses to manage the pandemic and other factors that can influence whether individuals experience severe COVID-19 outcomes, the International COVID-19 Policies and Epidemiology Working Group seeks to study the COVID-19 policy responses in jurisdictions across the world. This group aims to understand which physical distancing policies work and where, by situating country-specific COVID-19 policy responses in their broader geographical, political, and health systems context [8]. This working group has created a harmonized database to compile data on COVID-19 policies, epidemiology, and contextual information that are being used to inform case studies on COVID-19 policy responses in different countries and multinational comparative research that can ultimately inform policy-making.

Earlier studies have looked quantitatively at national-level COVID-19 government policy responses and the effects on COVID-19 epidemiology. One study looked at national-level physical distancing policies including lockdown, curfew and stay at home orders across 54 countries up until April 2020 and found that decreases in daily new cases of COVID-19 were associated with policy implementation [9]. Another study looked quantitatively at a larger range of physical distancing policies including school and workplace closures, limits on gatherings, public transport closures, and lockdown and their effects on COVID-19 incidence in 149 countries up until May 2020 [10]. The study found that implementation of physical distancing policies was associated with a reduction of COVID-19 incidence by an average of 13% and that earlier implementation was associated with a larger decrease in COVID-19 incidence. However, these studies did not describe the policies qualitatively in terms of start and end dates or in implementation across jurisdictions. Both of these studies also focused only on policy responses during the first wave of COVID-19. It therefore remains unclear what factors influence the success of physical distancing policies over time.

There is some comparative literature exploring how context has informed COVID-19 policy responses. One study compared public health and social measures used to manage COVID-19 in 10 countries, including China, Singapore, Japan, South Korea, UK, Italy, Germany, Sweden, US, and South Africa [11]. This study looked qualitatively at early COVID-19 policy responses, including physical distancing measures, and compared responses across jurisdictions. This study situated the COVID-19 policy responses in the broader political systems in these countries and found that these countries varied in timing and strictness of their policy responses depending on their political systems. However, this study focused only on upper-middle and high-income countries. Another study focused on comparing early COVID-19 policy responses in select countries within Africa in the context of their health systems [12]. The study found that early implementation of policy responses is important for pandemic management and that past experience with epidemics may be helpful with the COVID-19 response. However, this study did not look at a wide variety of physical distancing policies

in the selected countries or describe their implementation in detail. Given the limited exploration of COVID-19 policy responses in the context of low- and middle-income countries over time, it is worth researching jurisdictions in Africa to explore the impact of these physical distancing policies on COVID-19 epidemiology throughout the pandemic thus far.

Policy responses have been particularly important in Africa to prevent health systems from being overloaded. The Africa Center for Disease Control (CDC) led a response against COVID-19 that mobilized countries across the entire continent. The African Task Force for Coronavirus was created as a collaboration between Africa CDC, African Union members, and WHO to create policy recommendations to combat COVID-19 across the continent [13]. Despite coordinated efforts across the African continent to address the pandemic, there is variation in how African countries have fared during the COVID-19 pandemic [13]. African countries have also remained vulnerable to the effects of COVID-19 due to ongoing vaccine inequity [14].

South Africa became the epicenter of COVID-19 in Africa at the end of March 2020, which prompted a national lockdown [15]. The lockdown was initially successful in bringing down cases; however, South Africa eased restrictions due to economic concerns, even as COVID-19 cases continued to rise [16, 17]. In contrast, Uganda was celebrated for its initial COVID-19 response and was among the more successful African nations in curbing the spread of the virus [18]. Perceptions around these countries' track records in responding to COVID-19 set up interesting comparisons of how the COVID-19 policies have differed between these jurisdictions. It is worth exploring what policy decisions in these two countries have led to potentially differential outcomes with respect to COVID-19 epidemiology.

South Africa and Uganda also have interesting contextual factors that justify their comparison. Both are large, relatively stable African countries with democratic governments. Both have somewhat similar population sizes with recent experiences with infectious diseases and are rated similarly in terms of epidemic preparedness [19, 20]. South Africa is an upper middle-income country, while Uganda is considered a low-income country [21]. These differing income classifications affect the resources they have available for responding to a pandemic or providing socioeconomic support to their populations. Both countries are also English-speaking and collected and publicly reported COVID-19 data on a regular basis, which were important logistical considerations for conducting this research.

This study aims to answer the following research questions:

- What national-level physical distancing policies were enacted in South Africa and Uganda from January 2020 until November 2021, and how did these policies affect local COVID-19 epidemiology during this time?

- How did these policy responses compare across these jurisdictions?

The objectives of this study are to describe and understand the rationale behind the COVID-19 policy responses in South Africa and Uganda, the effects of these policy responses on the epidemiology of COVID-19, and how contextual factors may have affected COVID-19 policy development and timing.

## Methods

### Study design and case definition

This study followed a qualitative embedded multiple case study design [22]. Case studies can be used to answer "how" or "why" questions, and to explore current events in real-life contexts or events that are not under the control of the investigator. The case study design is being used

to look at COVID-19 physical distancing policy responses and their effects on local COVID-19 epidemiology in different countries as part of an ongoing larger multinational study. Each case was defined as the COVID-19 policy responses and resulting changes in COVID-19 epidemiology occurring in a specific country. South Africa and Uganda are the cases for this study. Cases were bound both by the country context and time, from January 2020, when the World Health Organization (WHO) declared a Public Health Emergency of International concern, until November 2021 when the delta waves had subsided in each jurisdiction [23]. This timeframe enabled analysis of key policy decisions and their subsequent effects on COVID-19 epidemiology within each country over multiple COVID-19 waves.

## Settings and participants

South Africa and Uganda are the settings for this study. Each setting was described in terms of geography and other important contextual factors. Contextual information was used to support interpretation of the findings and to assess potential transferability i.e., the extent to which findings can be applied to other settings.

Key informant interviews were conducted with participants such as policy makers, public health officials, and researchers from each jurisdiction to ensure accuracy and completeness of data collected, as well as to identify additional relevant data sources (e.g., other potential key informants, government documents).

## Participant sampling and recruitment

Personalized emails were used to contact potential key informants within each jurisdiction. Criterion sampling was used to select participants from each jurisdiction who were familiar with the policy-making process and factors that could influence policy development. Participants invited for interviews included policy makers, public health officials, and researchers. Snowball sampling was also conducted, whereby key informants were asked to identify other potential individuals knowledgeable on the research topic who could be contacted for an interview. The intention was to interview an estimated 1–5 individuals for each country to supplement findings from the documentary review.

## Data collection

Data were collected by a public health graduate student (SM) from July 2021 to April 2022 using a standardized data collection form for each country (S1 Appendix). Data were collected on country characteristics (geography, environment, social, economic, demographic, and health indicators) and political and health systems characteristics (e.g., type of government, health system financing) to help set the context for the pandemic response in each jurisdiction. Indicators were selected a priori and determined based on relevance, availability, and international comparability. Some measures were more direct (e.g., population ages) while others were used as proxy indicators (e.g., HIV, diabetes, and obesity prevalence as levels of immuno-compromise in a population). Data on country characteristics were drawn from public sources such as the World Health Organization (WHO), the World Bank, and the Central Intelligence Agency. Data were also collected on the overarching approach to COVID-19 management (e.g., containment, mitigation, herd immunity), physical distancing policies, and related supporting policies such as economic relief programs. Data on policy responses and contextual factors were collected from government websites, government reports, news articles, and peer-reviewed journal articles. Generic search terms such as "COVID-19 policy response", "COVID-19 media address", "COVID-19 physical distancing", and "COVID-19 lockdown" were used for each jurisdiction initially. Data collection on policy responses entailed searching

national government websites that had publicly available repositories of documents such as presidential addresses and media statements from cabinet ministers announcing COVID-19 measures. These repositories were scanned for COVID-19 relevant documents based on dates from January 1, 2020 to November 30, 2021. Research was conducted from Canada, therefore, to ensure that results pertained directly to each country's COVID-19 response, site-specific Google searches of South African and Ugandan news outlet websites were done to obtain information on specific types of policy responses (e.g., "South Africa school closures" or "Uganda public transport closures") where government announcements were unclear.

Sources specific to national-level COVID-19 responses in each jurisdiction were included in the documentary review. Documents explaining only sub-national (e.g., provincial) or local (e.g., municipal, village) level COVID-19 responses were excluded from the documentary review.

Epidemiological data were collected from publicly available sources such as Our World in Data. Policy decisions were explained in the context of timelines (e.g., first case, 100th case) and changing COVID-19 epidemiology (e.g., total number of cases and deaths in the country).

Following the documentary review, key informant interviews were conducted to confirm findings and cover any missing information. Recruitment took place from February 1, 2022 to March 16, 2022. A standardized semi-structured interview guide was used for each interview (S2 Appendix). English-language interviews were conducted remotely, using Zoom (version 5.9.3) [24]. Notes were taken during interviews. Interviews were recorded and transcribed with permission from participants to ensure accuracy. Audio recordings were produced using Zoom. SM created verbatim transcripts from the recordings. All data were stored securely on local devices that were password protected. Key informants were asked to provide documents which may be relevant to understanding events surrounding the pandemic. The primary goal of the key informant interviews was to supplement data collection from the documentary review.

## Data analysis

Data on country characteristics, political and health systems, and COVID-19 policies were input into a standardized data collection form for each country (S1 Appendix). Policy responses were documented according to type (overarching response strategy, physical distancing, vaccination, and socioeconomic support policies), start and end dates, descriptions of the policy implemented, and modifications over time. Qualitative analyses involved content analysis of resources included in the documentary review, following a deductive approach guided by the data collection form. Content analysis of notes and transcripts from key informant interviews were also used to provide supplementary information to the documentary review. A policy-relevant database of contextual factors and policies was created and is shared publicly on the COVID-19 Policies and Epidemiology Working Group website: https://covid19-policies.healthsci.mcmaster.ca/ In addition, for each country, a narrative summary was constructed into a case report to describe contextual factors, key policy decisions, and epidemiological events over the study period.

A comparative analysis was done using select contextual factors and major policies to explore similarities and differences in the COVID-19 responses and outcomes in South Africa and Uganda. Similarities and differences between the contextual factors and COVID-19 response policies implemented in each country were noted throughout the content analysis process. Tables with various contextual indicators were developed to compare contextual factors for each country. Policy responses for each jurisdiction were graphed against COVID-19 cases and deaths, compared to each other to observe similarities and differences in timing of

the policies, and used to identify any policy measures which appeared to curtail COVID-19 waves. Narrative summaries of the policy responses in the case reports were used to compare the nuances of policy implementation in each jurisdiction and other important factors in the response such as policy enforcement and public compliance.

### Knowledge translation

Detailed case reports documenting the South African and Ugandan COVID-19 policy responses were produced [25, 26]. Each case report is available online at: https://covid19-policies.healthsci.mcmaster.ca/research/publications/ This paper provides high-level overviews of the COVID-19 policy responses in each of these jurisdictions; however, its main focus is to report the findings of the comparative analysis of select key policies that were implemented.

### Ethics

This research is part of a larger multinational comparative study of physical distancing policies and their effects on local COVID-19 epidemiology, which received ethical approval in Canada from the Hamilton Integrated Research Ethics Board (HIREB) (Project # 11243). Key informants from South Africa and Uganda who were recruited for the study were informed of the purpose of the study, data confidentiality, and their right to participate voluntarily or withdraw their data at any point in the study until data was analyzed. To protect key informants who were potentially unable to speak freely about government actions and to avoid burdening participants who may have had limited access to technology (e.g., software for virtual signatures or scanners/printers), verbal consent was obtained from key informants prior to conducting the interview. Key informants responded verbally to consent questions and their responses were recorded by the public health graduate student (SM). Privacy was maintained by using study ID numbers instead of names for key informants in the case reports and comparative analysis.

**Inclusivity in global research.** Additional information regarding the ethical, cultural, and scientific considerations specific to inclusivity in global research is included in the Supporting Information (S1 Checklist).

## Results

### Data collected

**Documentary review.** A variety of sources were consulted for this documentary review including government websites, media reports, social media sites, peer-reviewed journal articles, and non-governmental organization (NGO) websites. No unpublished documents relevant to understanding events surrounding the pandemic were provided through the key informant interviews. Table 1 provides a breakdown of the number of each type of source that was reviewed.

**Key informant interviews.** A total of 10 potential key informants from South Africa were contacted for interviews, while 5 were approached from Uganda. In total, 6 key informants (3 from each country) agreed to be interviewed. Key informants were involved in the COVID-19 response in advisory and operational capacities within their respective jurisdictions. Key informants included a researcher and 5 public health officials. Interviews ranged from 22 to 47 minutes. Median interview length was 39 minutes, while mean length was 37 minutes.

**Table 1. Sources used in documentary review for each case study.**

| Type of Source | Number used for South Africa Case Study | Number used for Uganda Case Study |
|---|---|---|
| Government Sources | 91 | 33 |
| Media Reports | 14 | 44 |
| Social Media[a] | 0 | 5 (Ministry of Health Twitter, President's Twitter, Nation Television (NTV) News YouTube, Baba TV Uganda YouTube, Uganda Broadcasting Corporation (UBC) Television Uganda YouTube) |
| Peer-reviewed Journal Articles | 1 | 2 |
| Non-governmental organization (NGO) websites | 12 | 1 |
| **Total** | **118** | **85*** |

[a]Social media had multiple items from each channel

## Select contextual factors of South Africa and Uganda

It is important to understand the contexts within which the physical distancing policies were enacted. Several relevant contextual features are highlighted below. In-depth descriptions of Uganda's and South Africa's contextual features are provided in their respective case reports [25, 26]. Table 2 summarises these contextual factors.

**Table 2. Contextual factors.**

| | South Africa | Uganda |
|---|---|---|
| Population size, 2020 [19] | 59,308,690 | 45,741,000 |
| Population density, 2020 (people per square km of land) [27] | 48.9 | 228.1 |
| Population living in urban areas (%) [28] | 67 | 25 |
| Trust in national government (% of population) [29] | 42 | 55 |
| Global Health Security Index, 2019 (Overall Index Score out of 100 and category) [20][a] | 54.8–More Prepared | 44.3–More Prepared |
| Global Health Security Index, 2019 (Rapid Response Score out of 100 and category) [20][b] | 71.5–Most Prepared | 56.5–More Prepared |
| World Bank classification, 2020 [21] | Upper-middle income | Low income |
| Gini Index (score out of 100 and rank) [30][c] | Score: 63 (2014) Rank: 1 (2014) | Score: 42.8 Rank: 44 (2016) |
| Current health expenditure, 2019 (% of GDP) [31] | 9 | 4 |
| Physician density (physician per 1000 population) [32] | 0.8 (2019) | 0.2 (2020) |

[a] The Global Health Security (GHS) Index measures the ability of countries to prepare for and respond to infectious disease threats. The GHS Index provides an overall score, as well as scores across constituent categories. Scores are presented on a scale of 0 to 100, where 100 represents the best possible health security conditions. Scores are grouped into the following categories: "least prepared", "more prepared", and "most prepared".

[b] The rapid response category of the GHS Index assesses the ability to respond quickly to control epidemic spread.

[c] The Gini Index is used to quantify the level of income or wealth inequality in a jurisdiction. A number closer to 100 indicates perfect inequality.

**Geographic characteristics.** South Africa and Uganda have comparable population sizes. Population size and density can influence how quickly illness is able to spread amongst a population.

**Political and health systems characteristics.** South Africa is a constitutional democracy with three levels of government (national, provincial, and local) [33]. The African National Congress (ANC) is South Africa's current governing party and is led by President Cyril Ramaphosa, who has been in power since February 2018 [34]. The National Health Act (NHA) 2003 dictates the responsibility for health at each level of government [35].

Uganda's government is a presidential republic, meaning that the President of Uganda is the Head of State and Head of Government [36]. The National Resistance Movement (NRM) is the current party in power and is led by President Yoweri Museveni, who is serving his 6th term in office [37]. President Museveni has been in power since January 1986 and was re-elected on January 14, 2021. Uganda's health system is governed at the national and district levels [38]. In South Africa, a smaller proportion of people indicate trust in national government than in Uganda (Table 2).

South Africa and Uganda both have a mix of public and private healthcare. In each jurisdiction, there is a referral system for public health care delivery. South Africa has primary healthcare facilities, from where patients may be referred to district level hospitals to undergo more sophisticated testing or minor procedures, and tertiary hospitals where patients may be referred for major surgeries or specialized care [39]. In Uganda, public healthcare is provided through district level health services, regional referral hospitals, and national referral hospitals [40]. South Africa has a higher physician density and spends a higher percentage of their GDP on healthcare than Uganda (Table 2).

Both countries also have a mixed public and private lab system for testing. In South Africa, the National Health Laboratory System (NHLS) provides laboratory services for public healthcare support of the provincial and national health departments [41]. In Uganda, public laboratory services are provided by the Central Public Health Laboratories [42]. Like the public healthcare system, Uganda's laboratories have a tiered system where laboratories are housed at district health facilities, regional referral hospitals, and national referral hospitals.

**Epidemic preparedness and experience.** South Africa and Uganda had similar ratings of epidemic response preparedness (Table 2). However, South Africa and Uganda have different experiences dealing with infectious diseases. South Africa has dealt primarily with HIV and tuberculosis (TB). South Africa has a National Strategic Plan on HIV, TB, and sexually transmitted infections (STIs) 2017–2022 to address these diseases. Uganda deals with a variety of infectious diseases on a regular basis. They have past experience managing HIV, Crimean Congo hemorrhagic fever, Marburg virus disease, Rift Valley Fever, Measles, Cholera, and Ebola, among others. In 2018–2019, the neighbouring Democratic Republic of Congo had an outbreak of Ebola, which threatened to spillover into Uganda. As a result, the Ugandan Ministry of Health prepared by improving surveillance systems, risk communication, and Ebola prevention and treatment capacity [43]. These systems were in place and adapted for the COVID-19 response, as confirmed by a key informant.

**Income level.** South Africa is classified by the World Bank as an upper-middle income country, while Uganda is classified as a low-income country [21]. Both countries have had to mobilize financial and other resources to support their COVID-19 response. South Africa created a COVID-19 Solidarity Fund which raised money from individuals and businesses to support the government in buying items such as test kits and ventilators that were needed for the COVID-19 response, in addition to providing relief funds for vulnerable households [44]. Similarly, Uganda called on the public to donate food for vulnerable people and money to support the COVID-19 response [45].

**Table 3. Population age and health characteristics.**

| | | South Africa | Uganda |
|---|---|---|---|
| Population ages 0–14, total, 2020 (% of total population) [48, 49] | | 17,081,570 (29) | 21,048,125 (46) |
| Population ages 15–64, total, 2020 (% of total population) [50, 51] | | 38,959,544 (66) | 23,784,596 (52) |
| Population ages 65 and above, total, 2020 (% of total population) [52, 53] | | 3,267,576 (6) | 908,279 (2) |
| Current tobacco use prevalence, 2018 (%) [54] | Total | 31 | 10 |
| | Male | 47 | 16 |
| | Female | 16 | 4 |
| Prevalence of obesity among adults (Body Mass Index ≥30), 2016 (%) [55] | Total | 27 | 4 |
| | Male | 15 | 2 |
| | Female | 39 | 7 |
| Prevalence of Human Immunodeficiency Virus (HIV) (% of population ages 15–49), 2020 [56] | | 19 | 5 |

**Inequality.** Both countries are rated high on the Gini Index (Table 2). South Africa had the highest income inequality in the world in 2014 [30]. As of 2016, Uganda ranked 44th in the world. The inequalities in both of these jurisdictions have had implications for vulnerable populations experiencing disproportionate harms from COVID-19 policy responses, such as women who were affected by an increase in gender-based violence during lockdown [46, 47].

**Population age and health characteristics.** Both South African and Ugandan populations are notably young (Table 3). The high proportion of young people in these countries may have implications for COVID-19 symptom presentation, especially in Uganda. The age distribution of these populations is also important to consider in the context of school closures.

Certain population health characteristics including smoking, obesity, and HIV prevalence may have implications for COVID-19 severity (Table 3). In South Africa, these indicators are higher than in Uganda.

## COVID-19 responses

The COVID-19 policy responses in South Africa and Uganda can be compared on the basis of select major physical distancing policies including lockdown, school closures, international travel bans, public transportation measures, curfew, and quarantine. Their responses can also be compared in terms of relief programs used to support the population with physical distancing, vaccinations, and enforcement of policies. Figs 1 and 2 show cases, deaths, and a timeline of select COVID-19 physical distancing policies in South Africa and Uganda respectively [57].

**Overarching COVID-19 response strategies in South Africa and Uganda.** The COVID-19 policy response decisions in South Africa were made by the National Coronavirus Command Council (NCCC), which was chaired by President Cyril Ramaphosa and composed of federal cabinet members. South Africa's first COVID-19 case was detected on March 5, 2020 and this number had risen to 61 by March 15, 2020 [58, 59]. At that time, South Africa declared a National State of Disaster and began implementing public health measures in response, including a travel ban, gathering limits and school closures. Following a lockdown beginning March 26, 2020, South Africa's pandemic response was marked by lifting and reinstating public health measures depending on COVID-19 transmission and hospital capacity, which was somewhat guided by a 5- level alert system. Alert Level 5 was a full national lockdown, while lower levels had progressively relaxed measures [60].

Uganda's COVID-19 policy response decisions were made by their National Taskforce on Coronavirus, which was led by President Yoweri Museveni [61]. Unlike South Africa, Uganda

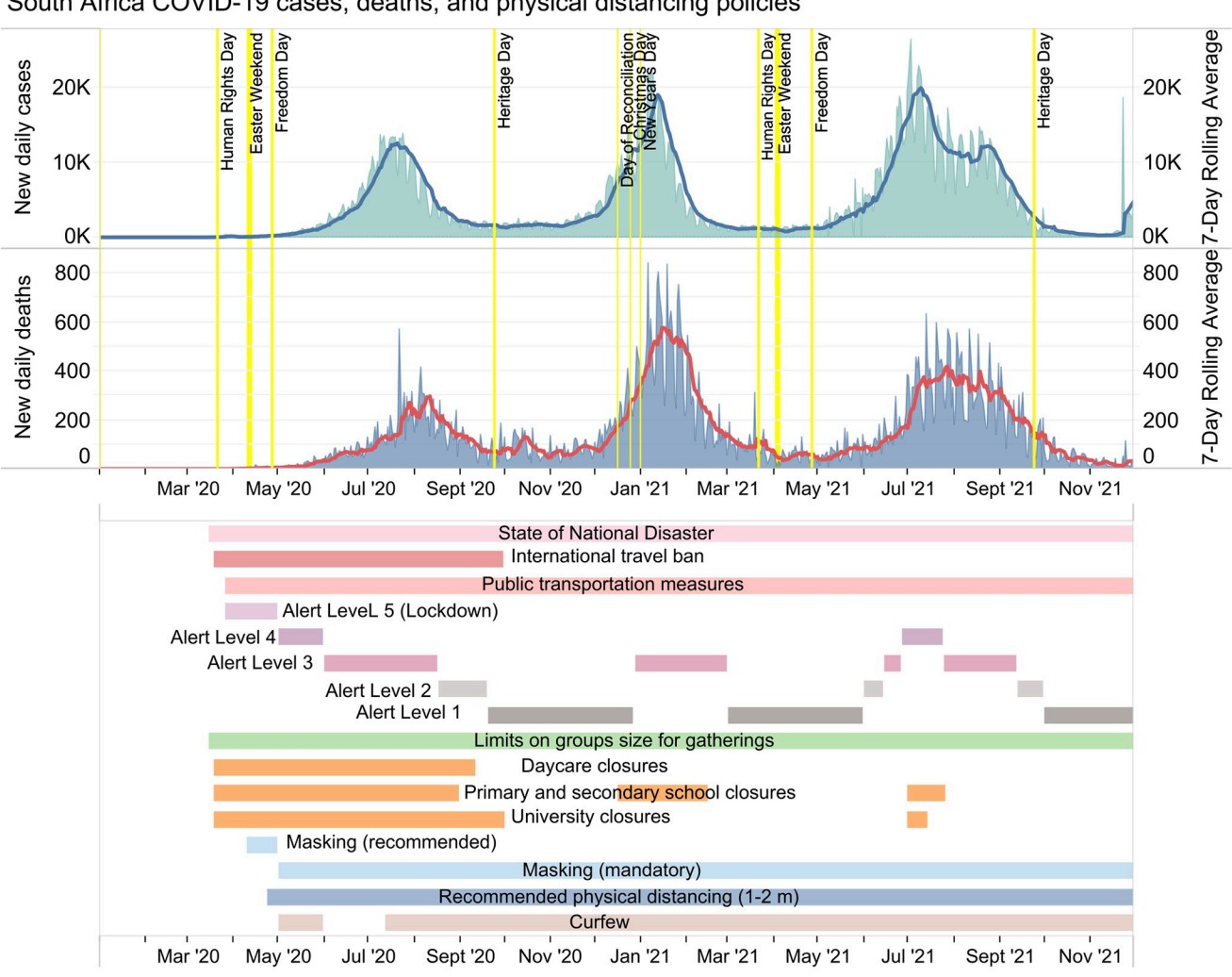

**Fig 1. South Africa's COVID-19 cases, deaths, and timeline of key policies.** Alert Levels 1–4 depended on rising cases and hospital capacity and had implications for gathering limits and curfew hours; however, there was no published threshold for each alert level, and policies were nuanced and changed over time.

never officially declared a National State of Emergency. Instead, a key informant explained how they used presidential directives:

> "What we had were presidential directives. Presidential directives were then sort of codified in law, through what we call COVID-19 rules, essentially laws, like many amendments to our Public Health Act, such that once the President made directives on travel, or curfews or business operation, the Ministry of Health's legal team would then work with the Ministry of Justice (. . .) and then that would then be legally binding."

–Interview 001 (Uganda, public health official)

Uganda's COVID-19 response began prior to detecting their first case. Screening measures were implemented at Entebbe International Airport beginning in January 2020, before WHO declared COVID-19 a public health emergency [62]. Uganda's Ministry of Health created the

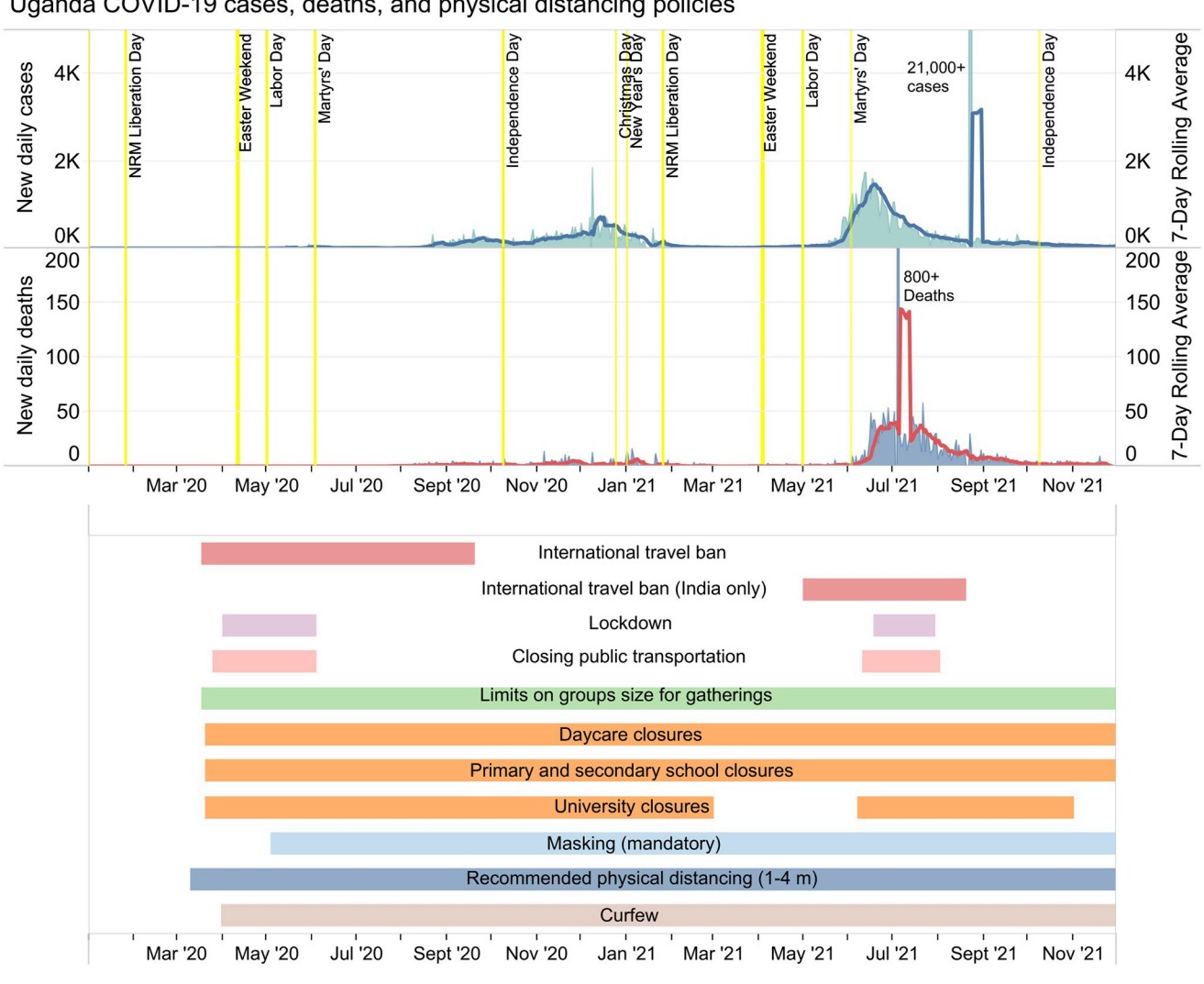

**Fig 2. Uganda's COVID-19 cases, deaths, and timeline of key policies.**

National COVID-19 Response Plan for March 2020 to June 2021 [61]. The plan was aimed at reducing COVID-19 entry into the country, transmission, morbidity, and mortality while also minimizing the socioeconomic ramifications from the pandemic. Uganda later developed a National COVID-19 Resurgence Plan June 2021 –June 2022 that was meant to build off the original response plan and help deal with scenarios where cases were rising again [63].

COVID-19 responses can be categorized as containment or mitigation approaches [64, 65]. A containment approach to COVID-19 policy is one that aspires to zero community transmission. By contrast, a mitigation approach seeks only to reduce COVID-19 transmission to the extent necessary to maintain societal function while preventing collapse of the healthcare system.

South Africa's COVID-19 response can be defined as a mitigation approach. A key informant described South Africa's approach as follows:

"We never at any stage opted for zero COVID, or total containment, rather, we went for minimizing the number of cases, and our stated goal was to flatten the curve. So largely, it was about slowing transmission, and mitigating the impact."

–Interview 002 (South Africa, public health official)

Uganda's COVID-19 response began initially as a containment approach which aimed to use travel-related policies to prevent COVID-19 from entering the country and reaching the community. Uganda later switched to a mitigation strategy, which coincided with the lifting of lockdown measures and rise in community transmission [66]. A key informant described Uganda's approach as follows:

"So the initial phase, containment was the strategy, it was the goal. However, community spread started around, you know, July, August, there was evidence now of community spread at that point. . .So, I would say it's, it's more or less a story of, you know, two approaches, one for containment. But a second more comprehensive plan once it became clear that community transmission had been established, where the focus was more on management of cases, infection control and health facilities, community-based surveillance, and ramping up of testing."

–Interview 001 (Uganda, public health official)

**Lockdown.**   Both jurisdictions implemented lockdowns that required individuals to stay home except for essential reasons and involved widespread closures of non-essential services across different economic sectors. South Africa had only one national lockdown at the beginning which lasted approximately 4–5 weeks. Uganda also implemented a full national lockdown at the beginning of the pandemic; however, it was sustained for a period of approximately 2 months.

Both countries phased out their lockdowns progressively. Lifting of lockdowns was related to cases subsequently rising in both jurisdictions. In South Africa, lockdown measures were lifted according to the risk-adjusted measures framework, where lower alert levels signalled a relaxation in public health measures [60]. Uganda did not have a particular published framework for lifting their measures. While South Africa had a risk-adjusted measures plan that was used throughout the pandemic, the measures that each alert level implied were not kept consistent over time. Despite their risk-adjusted plan, South Africa's approach approximated Uganda's in the sense that public health measures were announced and implemented as the government deemed necessary.

South Africa never returned to a full national lockdown for subsequent COVID-19 waves, after their initial lockdown during the first wave. The closest they came to a second lockdown was during their third wave as the delta variant was taking off, moving up to Alert Level 3, and then up to Alert Level 4 as cases continued to rise even further. In comparison, Uganda did commit to a second lockdown during their second wave, which was driven by the delta variant [67]. The second lockdown appears to have played a meaningful role in bringing case numbers down, based on the decline in cases that followed the implementation of the lockdown (Fig 2).

**International travel bans.**   Both countries also initially attempted to prevent COVID-19 cases from entering by implementing international travel bans from high-risk countries. South Africa also initially restricted travel into South Africa for foreign nationals from high-risk countries, and by closing 35 of 53 land ports and 2 of 8 seaports [59]. Once the lockdown went into effect on March 26, 2020, these measures were made stricter. All international flights,

cross-border road transportation from neighbouring countries, and passenger ships were prohibited from entering South Africa [68]. These measures remained in place until October 2020 and all international flights and cross-border road transportation were permitted [69]. South Africa's international travel ban only applied during the first wave of COVID-19. Borders remained open during subsequent waves, subject to requirements around quarantine and testing.

Uganda's approach involved issuing a travel ban specifically for countries with a large number of COVID-19 cases, which they later followed up with a complete border closure whereby all entry into Uganda by air, land or water was prohibited [70, 71]. This ban prohibited buses, mini-buses, and commercial motorcycles from entering Uganda from neighbouring countries, in addition to pedestrians and cyclists [71]. These measures were enforced by Ugandan security forces. Border measures played an important role in Uganda's containment approach as they focused initially on preventing importation of any COVID-19 cases. Later on, however, borders reopened in September 2020 for tourism [72]. During the second wave, there was much less emphasis on preventing cases from being imported. Instead, the travel ban was once again reserved for countries categorized as high risk by Uganda, which only applied to India during the delta wave [73]. All flights to India from Uganda were prohibited, and travellers arriving from India or who had travelled through India in the past 14 days were barred from entering Uganda.

**Public transportation.** In South Africa, public transportation never fully shut down and their policies varied across different modes of transportation and based on distance travelled. During the lockdown, public transportation involved some closures including domestic flights and passenger rail operations [74]. Road transit including minibus taxis, metered taxis, and e-hailing services were only allowed to transport essential workers during specific hours. They had also set capacity limits based on the licensed maximum for different vehicle types. In the case of minibus taxis and buses, a 70% capacity limit was set, whereas for metered taxis and e-hailing services it was only 50% [74]. Public transportation measures were eased progressively alongside the lockdown to support commuters who were being allowed to return to work [75]. By waves 2 and 3, eventually the main measure that remained was a 70% capacity limit that applied only to public transportation vehicles travelling long distances (defined as 200 km or more) [76–84]. Local public transportation trips could carry 100% of licensed capacity provided that everyone was wearing a mask.

Uganda shut down their public transportation, which coincided closely with their lockdowns. Initially, as Uganda started implementing proactive measures prior to case detection, they recommended that individuals only use public transportation for essential reasons [70]. During the first lockdown, they instituted a ban across several modes of transportation including taxis, minibuses, buses, trains, and commercial motorcycles [85]. Once the closure had ended, they implemented 50% capacity limits for minibuses, buses, and taxis [86]. During the second wave, the public transport closures were more relaxed than during the first wave. Public transport including buses, taxis, and commercial motorcycles could not be used to travel between districts, but they could be used to travel locally within districts [87].

**School closures.** South Africa and Uganda's approaches to school closures were also noticeably different. In South Africa, schools were initially closed during the lockdown and throughout wave 1; however, they underwent recurrent openings and closures in subsequent waves. While it is not possible to pinpoint exactly which policy decisions were most useful in mitigating COVID-19 transmission, it is interesting to note that South Africa's closure of primary and secondary schools aligned most closely with reducing cases during each COVID-19 wave South Africa experienced (Fig 1). In Uganda, school closures began prior to the country's first recorded COVID-19 case and were in effect for close to 2 years for most students [70].

While it is challenging to estimate exactly how effective school closures were in Uganda, it is possible that the sustained closures over 2 years contributed to lower COVID-19 transmission than it otherwise could have been given that approximately half the population of Uganda is 17 years of age or younger, and that this age group remained ineligible for vaccination over the study period.

**Gathering limits.** Both governments introduced limits on gathering size, which fluctuated throughout the pandemic. Gathering limits were changed regularly in South Africa, ranging from outright bans, and going up as high as 750 people indoors or 2000 people outside [88]. Uganda's approach to gathering limits was somewhat more conservative than South Africa. While they also varied over time, they only went as high as 200 people, indoors or outdoors [89].

**Curfew.** A curfew was implemented in both countries; however, they were guided by different rationales. In South Africa, curfew was instituted to help preserve the healthcare system by reducing accidents overnight, thereby limiting the need to divert healthcare resources towards trauma-related care [90]. Curfews were among the policies that remained most consistently implemented in South Africa. In Uganda, curfew was implemented to promote physical distancing. The reasoning was that many services would have to close due to curfew, therefore reducing the places where people could congregate, and prompting people to remain at home for the night [91].

**Quarantine for travellers, confirmed cases, and contacts.** South Africa and Uganda had similar quarantine policies. Both countries required a 14-day quarantine following travel, and for confirmed cases and contacts starting in March 2020 [68, 70]. South Africa reduced this time from 14 days to 10 days in July 2020; Uganda did so in late September 2020 [92, 93]. In October 2020, both countries changed their policies for travellers by exempting individuals who could present a negative COVID-19 test from quarantine. Individuals unable to present a negative test result were still required to quarantine for 14 days until they tested negative [93, 94].

**Vaccination.** Both South Africa and Uganda began efforts to procure vaccines in December 2020 through the WHO COVAX facility, Africa Union Vaccine Initiative, and direct purchase from manufacturers [95]. South Africa paid R283 million (18,082,814 USD) towards the COVAX facility to secure vaccine doses for their population. South Africa was expecting to receive enough doses to vaccinate 10% of the population in 2021 [96]. South Africa received its first shipment of AstraZeneca vaccines in early February 2021; however, given a study showing that AstraZeneca had reduced effectiveness against the beta variant, South Africa delayed their rollout until they received Johnson & Johnson and Pfizer vaccines instead [97].

While Uganda also began efforts to procure vaccines in December 2020, health experts were concerned that the government lacked a clear plan or budget for vaccine procurement, which would affect Uganda's ability to access vaccines in a timely manner [98]. Like South Africa, Uganda also applied to the WHO's COVAX facility for vaccinations [99]. The COVAX facility was expected to support vaccination of 20% of the population. Uganda received donations of Astra-Zeneca from the Indian government and CoronaVac (Sinovac) vaccines from the Chinese government [100, 101]. Initially, Uganda chose to begin their vaccination campaign with Astra-Zeneca and Chinese vaccines because they were easier to store than mRNA vaccines [102].

Both countries also had similar strategies for prioritizing who would be eligible for vaccination. In South Africa, the vaccination campaign was divided into three phases. Phase 1 began with healthcare workers. Phase 2 focused on frontline workers such as teachers and police forces, people living in congregate settings including old age homes, shelters, and prisons, people over the age of 60, and adults with comorbidities. Phase 3 was focused on the remaining

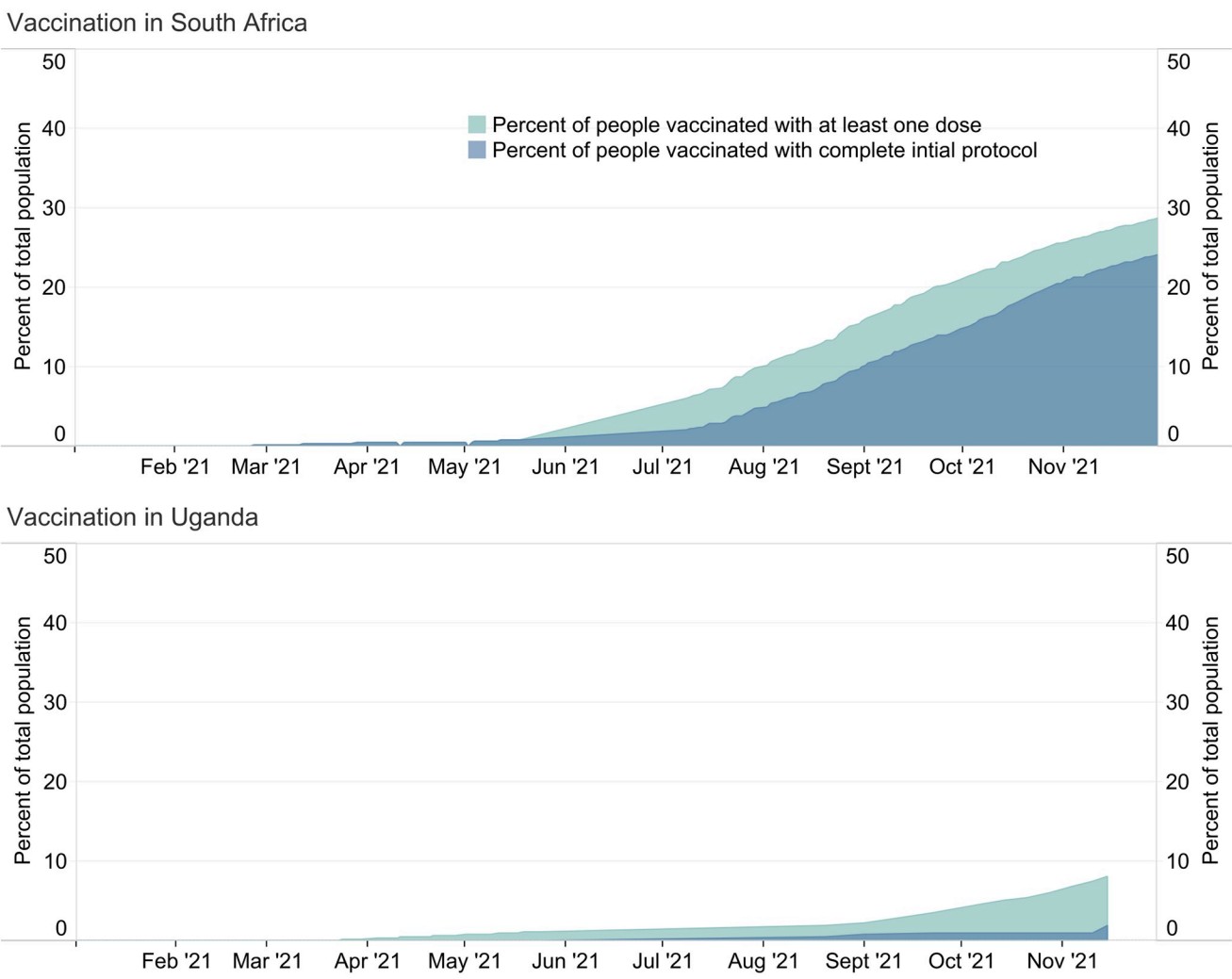

**Fig 3. Vaccination in South Africa and Uganda from January 2021 to November 2021.** People vaccinated with a complete initial protocol means they received all initially prescribed doses for the vaccine brand they received.

adult population [103]. Similarly, in Uganda, health care workers were the first group to be vaccinated to ensure they were protected from severe outcomes from COVID-19 and to encourage them to continue providing care to infected individuals [104]. Teachers were also prioritized for vaccination to facilitate safe school reopening. Adults over 50 and adults aged 18–50 with health conditions including hypertension, diabetes, cancer, and other diseases including liver, kidney, or heart disease were considered high-risk groups for contracting COVID-19 and were also prioritized for vaccination.

Vaccine rollout began in South Africa in mid-February 2021 [97]. By September 2021, South Africa had enough vaccine supply for their entire adult population [105]. In comparison, vaccination in Uganda began on March 10, 2021 [106]. Uganda had received enough vaccines for their adult population by November 2021 [107].

Vaccination progress remained slow in both jurisdictions by the end of the study period. Fig 3 shows the percentage of the total population that had been vaccinated with at least one dose or with a complete initial vaccine series by November 2021 for each jurisdiction [57].

Neither jurisdiction had particularly high vaccine coverage. In South Africa, close to 30% of the total population had received their first dose by mid-November 2021, whereas in Uganda it was especially low, at less than 10% (Fig 3).

In South Africa, low vaccine coverage was attributed to barriers to access such as hours of operation of vaccine clinics that coincided with business hours where people were working [108]. In response, the government expanded their vaccination so that individuals could get their vaccinations on select weekends. The government tried to make vaccines easier to access in areas where vaccine uptake was disproportionately low by making them available at polling stations during local elections [109]. The government also piloted an incentive program, whereby they would provide R100 grocery vouchers to people over the age of 60 who received their first dose in November 2021 [109]. A key informant also pointed to the issue of misinformation and vaccine hesitancy:

> "I mean, South Africans never had issues with vaccine hesitancy before. You know, I've worked in vaccine trials for many years. And it was never an issue, you know, people just vaccinated. But I think with, you know, the, there's been a lot of misinformation and I think with social media and things being freely available, a lot of that crept in."
>
> –Interview 003 (South Africa, researcher)

In Uganda, in early November 2021, the Ministry of Health had only begun expanding the vaccination program from health facilities to community locations including villages, churches, taxi parks, and markets to increase vaccine coverage in Uganda [110]. By the end of the study period, all adults over the age of 18 had just become eligible for vaccination [107]. However, children under the age of 17 were still not yet eligible to be vaccinated despite making up half of the total population. A key informant also cited vaccination misinformation on social media as a problem in Uganda:

> "Then of course the other problem was demystifying facts about the vaccine, the rumors around the vaccine. Countering anti-vaxxers, you know, there's so many things with the social media now."
>
> –Interview 004 (Uganda, public health official)

**Socioeconomic relief policies.**   Both the governments of South Africa and Uganda had instituted relief programs for populations that were vulnerable to the effects of public health measures to control COVID-19 transmission. The South African government provided financial and food support to their population. South Africa used the Unemployment Insurance Fund (UIF) to create the COVID-19 Temporary Employee Relief Scheme (COVID-19 TERS) [111]. The COVID-19 TERS benefit was provided to employers so that they could continue to pay employees their regular salary while they remained home during lockdown. The government also supplemented existing social grants such as child support [112]. A temporary COVID-19 Social Relief of Distress grant provided R350 (22.36 USD) per month to unemployed individuals or to individuals who were otherwise not receiving any social grants [112]. This grant was originally available from May-October 2020 and was extended periodically until it was paused in April 2021 because the government could no longer afford to provide it [113]. The government later reinstated the grant in August 2021. South Africa also had a food program that was coordinated by the national Ministry of Social Development [114].

Emergency food parcels were delivered to registered people through existing food programs, as well as people who were not already supported by the Social Relief of Distress grant.

During Uganda's first lockdown, they also introduced a food distribution program to support those experiencing food insecurity [115]. The government had limited means to provide support for the population during lockdown, therefore food support was reserved for people whose incomes were directly affected by government public health measures and who could also not grow their own food [116]. During the second lockdown, the Ugandan government took another approach to supporting the population, opting to provide cash transfers rather than food packages [117]. The cash transfer program provided UGX100,000 (26.48 USD) to vulnerable people; however, this program also had limited reach. More than 7.7 million people living in poverty were not supported by the program [118]. The government justified their choice of beneficiaries, claiming that the people not included in the program were not necessarily considered vulnerable even if they were poor, owing to them living in rural areas and being able to produce their own food.

**Public response and policy enforcement.**   The public reaction to COVID-19 measures in South Africa was mixed. A poll conducted in South African urban centres from April 2 and April 6, 2020 demonstrated that 83% of South Africans were satisfied with the government's response to COVID-19 [119]. A later poll conducted from April 20 to April 22, 2020 suggested that 84% of South Africans supported the lockdown, believing that the risk posed by the pandemic justified the response [120]. Despite indications of general support for the South African government's COVID-19 response, people with lower household income were less likely to be satisfied with the government response or to trust the information provided by the government [119].

In Uganda, a public opinion poll taken from July 6 until July 15, 2020 found that 75% of respondents initially approved the government's decision to implement a lockdown [121, 122]. However, up to 78% of the participants polled reported that they had stopped fearing COVID-19 and were less likely to follow public health measures.

Police forces were relied upon in both South Africa and Uganda to enforce public health measures. In South Africa, lockdown measures were enforced by the South African Police Service and the South African National Defence Force. Police conducted patrols to limit movement of individuals [123]. Violation of public health measures during lockdown could result in a fine, imprisonment, or both. Police involvement in South Africa during lockdown was controversial because of police violence against individuals who disobeyed public health regulations [124]. The South African government also criminalized violation of the mask mandate [95]. Failure to wear a mask in public spaces could result in a fine, prison time, or both.

In Uganda, police were also responsible for patrols during lockdown and ensuring individuals' movements were limited to essential reasons [45]. A key informant confirmed that the police played a major role in enforcing curfew, which was in effect for most of the study period. Police violence against individuals violating COVID-19 measures was also noted, which prompted the President to emphasize that police should arrest people going against measures rather than resorting to violence [45].

## Discussion

### Principal findings

This study is part of a larger multinational study of physical distancing policies and their effects on local COVID-19 epidemiology [8]. Within this project, we chose to study the COVID-19 policy responses of South Africa and Uganda in an effort to document African responses, while situating the policies in the broader context of each country.

The suite of physical distancing policies used in South Africa and Uganda to prevent COVID-19 transmission were similar in the sense that they both used lockdown, school closures, international travel bans, public transportation measures, gathering limits, curfews, and quarantine for travellers, confirmed cases, and contacts. However, policy implementation differed on several points. South Africa only began implementing public health measures after cases began rising rapidly. South Africa never aimed to achieve zero community transmission, aspiring instead to only mitigate spread. By comparison, Uganda's policies approach was more proactive than South Africa. Uganda began implementing policies prior to a single case being detected. Their initial approach was grounded in complete containment of the virus by preventing importation of cases from initiating community spread.

Mitigation strategies in both South Africa and Uganda's COVID-19 responses were marked by a need to preserve the economy. Despite Uganda's shift in strategy away from total containment of the virus, mitigation measures in both countries had a different character. South Africa's response was marked by a recurrent tightening and relaxation of public health measures over the course of the study period, depending on whether cases were rising and whether hospitals could respond to any changes. In their policy decisions, it seemed that the South African government wanted to ensure that some public health measures were in place to control transmission, but only to the extent that they felt was absolutely necessary. In comparison, Uganda sustained policy measures for longer periods of time, that were also stricter in nature, prior to lifting them. Based on several of Uganda's policies such as school and public transportation closures, longer quarantine periods, and their decision to reinstitute a national lockdown during the second wave, their government was willing to go further than South Africa with their measures to reduce the risk of COVID-19 transmission over the study period.

It is worth noting that South Africa had three recorded waves over the same time period in which Uganda had only two waves (Figs 1 and 2). COVID-19 responses in both countries have entailed a complex layering of physical distancing policies which overlapped in time and may individually have had varying degrees of effectiveness. While it is difficult to say exactly what role policy responses played in mitigating the number of waves in each country, it is still worth considering the differences between South Africa and Uganda's approaches in terms of timing and stringency of measures, and how these differences may have influenced the course of the pandemic in each of these jurisdictions. As of November 30, 2021, there were 49,432 cumulative cases per million people in South Africa and 2,706 cumulative cases per million people in Uganda [125]. At the same time, there were 1,496 cumulative deaths per million people in South Africa and 69 cumulative deaths per million in Uganda.

Country context may also have played a role in these outcomes. South Africa's population is more concentrated in urban centres in comparison to Uganda. South Africa and Uganda have young populations overall; however, Uganda's is especially notable with half the population being under 17, and not being able to congregate in schools. Both countries were similarly prepared to respond to epidemics; however, Uganda's recent experience with Ebola just prior to the emergence of COVID-19 may have helped them activate their response systems faster.

Both South Africa and Uganda provided relief measures to support vulnerable populations during the pandemic using a mix of strategies. South Africa's aimed mainly to use existing social service programs and supplement them, although they did also implement programs for people not otherwise receiving any support. Uganda's programs were introduced to provide support during the lockdowns. Both countries at times provided grants and food parcels to support vulnerable individuals and households. Both countries are resource-limited and raised money to support their COVID-19 responses. In these resource-limited settings with notable income and wealth inequality, it is important to note that attention was still given to providing socioeconomic supports for people most vulnerable to the effects of public health measures.

## Placing this study in the literature

Similar to our research group, others have also studied COVID-19 policy responses. The Cambridge Core blog looked at country responses to COVID-19, as well as the responses of some sub-national jurisdictions [126]. These reports provided high-level information on jurisdiction contexts and overviews of the policy responses focusing primarily on the first several months of the pandemic response. However, the Cambridge Core blog did not cover the COVID-19 responses of any African countries.

The Oxford COVID-19 Government Response Tracker (OxCGRT) also collected policy information about public health measures taken against COVID-19 since January 2020 [127]. They have developed a database of indicators on economic, health system, vaccine, and containment policies. The OxCGRT has a stringency index, which measures strictness of government policies according to the following areas: school closures, workplace closures, cancellation of public events, limits on public gatherings, public transportation closures, stay-at-home measures, public education campaigns, international travel measures, and limits on internal movements. The stringency index averages scores for each of these policy areas to create a composite score from 0 to 100, with 100 being the strictest. While the OxCGRT does not contextualize policy responses according to changing local epidemiology, the OxCGRT stringency index illustrates that South Africa's measures were at their strictest during their national lockdown, rating at 87.96, and never again reached the same strictness for the rest of the study period [128]. By comparison, it showed that Uganda's maximum stringency was 93.52 during their first lockdown, and that measures were similarly strict at 87.04 when their second lockdown was instated. These metrics align well with our findings.

The International Labour Organization (ILO) explored economic measures to support businesses, employees, and vulnerable populations against the effect of COVID-19 [129]. The ILO has provided information on economic policies up until June 2020 for Uganda and December 2020 for South Africa. The International Monetary Fund (IMF) has covered the COVID-19 policy responses in various countries including South Africa and Uganda up until July 1, 2021 [130]. The IMF provided overviews of physical distancing measures in South Africa and Uganda, which aligned well with our findings. Similar to the ILO, The IMF also had a particular focus on economic policies and documents these in more detail than our study. However, the IMF policy tracker did not incorporate other policy-relevant contextual information.

## Strengths and limitations of the study

The case study design used in this research allowed for an in-depth exploration of country-specific contextual factors and COVID-19 physical distancing and supporting policy responses in South Africa and Uganda over time. The study drew from a wide variety of sources including government and media reports, and key informant interviews to confirm findings.

Limitations to this study exist because findings rely on availability of complete, accurate policy and epidemiological data being reported by the government, the media, and other publicly available sources. Differences in timeliness and reporting between countries are also important considerations when comparing COVID-19 policies and making inferences about their impact. State capacity to implement and enforce policies were not always clear, nor was public compliance, which may have affected our findings. In cases where policies were implemented later than announced or not implemented at all, it is difficult to assess their impact and there may be some discrepancies with the findings presented in this paper. A separate paper from the COVID-19 Policies and Epidemiology Working Group explores issues of policy implementation and limitations of publicly reported COVID-19 data [131]. Standardized

data collection forms and interview guides were used to conduct this research; however, the policy responses were nuanced for each country which made it challenging to ensure fidelity of the policy descriptions or key informant interviews. While English is an official language in both Uganda and South Africa, only English-language documents were analyzed for this study, which means that potentially important resources may have been missed. Only a small number of key informants from each country participated in interviews. The key informants were knowledgeable on the COVID-19 response in their respective jurisdictions; however, they may not have been involved in all aspects of decision-making or implementation.

## Implications for policy and practice

This paper has investigated the COVID-19 response beyond the first wave and explored how policy decision-making has differed over subsequent waves in South Africa and Uganda up until November 2021. Exploring the COVID-19 responses over time generates knowledge on how to deal with the ongoing nature of managing a pandemic in terms of implementing policy measures to mitigate or control the virus, vaccinating the population, handling emerging variants, and addressing socioeconomic challenges arising from public health policies.

Based on the COVID-19 responses in South Africa and Uganda, policy-makers and other stakeholders should consider their overall goals for pandemic management and think about how factors such as timing and stringency can affect the success of their pandemic management. They should also consider the impact of policy measures on vulnerable populations and find ways to provide socioeconomic support to these populations in the face of prolonged public health measures.

## Implications for research

While other work exploring COVID-19 policies has been done, many are focused on economic policies. To our knowledge, there are a limited number of studies conducting comparative analyses focused specifically on different types of physical distancing policies especially in countries within Africa. Continued exploration of pandemic management in African countries is needed to provide insight around decision-making in resource-limited settings.

Sources used for this study varied based on jurisdiction. South Africa's government had a complete repository of policy announcements published on their website and other resources were used primarily to gauge public response to mitigation measures. While Uganda's government websites had resources on their policy measures, other sources were needed to help contextualize their implementation. For Uganda, social media were particularly useful for mapping out policy timelines and locating presidential addresses. Future policy research should consider how social media sources may be used to supplement government web sources.

This paper used the Global Health Security (GHS) Index to understand pandemic preparedness in South Africa and Uganda. While their GHS Index scores were similar, the COVID-19 outcomes were different in these two countries [20]. This finding suggests that future research could focus on developing indicators to assess pandemic preparedness that consider the factors that contributed to successful COVID-19 national responses.

## Conclusion

This study focused on providing in-depth comparisons of COVID-19 policy responses and relevant contextual factors in South Africa and Uganda. The study showed how contextual factors such as population age, geographic distribution, and recent epidemic response experience can influence COVID-19 transmission and response. The study also showed differences in overall

strategy, timing, and strictness of epidemic management policies in these jurisdictions. These findings suggest it may be important to have sustained, strict measures to limit the spread of COVID-19 and manage the course of a pandemic, which need to be further explored alongside other important social and economic pandemic outcomes.

## Supporting information

**S1 Appendix. Standardized data collection form.**
(DOCX)

**S2 Appendix. Semi-structured interview guide.**
(DOCX)

**S1 Checklist. Inclusivity in global research checklist.**
(DOCX)

## Acknowledgments

We would like to acknowledge all key informants from South Africa and Uganda who participated in interviews, as well as individuals who were contacted for interview and connected us to alternative contacts who they believed could best support this work.

## Author Contributions

**Conceptualization:** Elizabeth Alvarez.

**Formal analysis:** Sana Mohammad.

**Investigation:** Sana Mohammad.

**Methodology:** Elizabeth Alvarez.

**Supervision:** Emma Apatu, Lydia Kapiriri, Elizabeth Alvarez.

**Visualization:** Sana Mohammad.

**Writing – original draft:** Sana Mohammad.

**Writing – review & editing:** Emma Apatu, Lydia Kapiriri, Elizabeth Alvarez.

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
