## [Decision Letter · Decision Letter 0]

19 Feb 2024

PGPH-D-23-02580

A comparative analysis of COVID-19 physical distancing policies in South Africa and Uganda

Dear Dr. Mohammad,

Thank you for submitting your manuscript to PLOS Global Public Health. After careful consideration, we feel that it has merit but does not fully meet PLOS Global Public Health’s publication criteria as it currently stands. Therefore, we invite you to submit a revised version of the manuscript that addresses the points raised during the review process.

We look forward to receiving your revised manuscript.

Kind regards,

Veena Sriram

Academic Editor

Journal Requirements:

Additional Editor Comments (if provided):

Reviewers' comments:

Reviewer's Responses to Questions

**Comments to the Author**

1. Does this manuscript meet PLOS Global Public Health’s publication criteria? Is the manuscript technically sound, and do the data support the conclusions? The manuscript must describe methodologically and ethically rigorous research with conclusions that are appropriately drawn based on the data presented.

Reviewer #1: Yes

Reviewer #2: Yes

Reviewer #3: Partly

2. Has the statistical analysis been performed appropriately and rigorously?

Reviewer #1: N/A

Reviewer #2: N/A

Reviewer #3: I don't know

3. Have the authors made all data underlying the findings in their manuscript fully available (please refer to the Data Availability Statement at the start of the manuscript PDF file)?

Reviewer #1: Yes

Reviewer #2: Yes

Reviewer #3: Yes

4. Is the manuscript presented in an intelligible fashion and written in standard English?

Reviewer #1: Yes

Reviewer #2: Yes

Reviewer #3: Yes

5. Review Comments to the Author

Reviewer #1: Thank you for an excellent research paper about the comparative CoViD response in South Africa and Uganda.

while I found the paper well researched and thorough, I feel that there are some areas where you could supplement the material in the paper

1. Figures 1 and 2 show that there were three waves in South Africa and only two in Uganda.. as a matter of fact, in Uganda, the first wave (which was much smaller than any of the waves in South Africa) started only in Oct 2020 unlike in South Africa where the first wave was around August 2020. Both the countries implemented a nationwide lockdown in March. Uganda extended its lockdown to around June, while South Africa relaxed the same in May. Both had international travel bans in place. It is not clear from the various policy measures taken why the epidemiological pattern was so different.. and how Uganda escaped the wave in August 2020 or why the number of cases in Uganda in both waves was so much lower(as was the mortality)

2.You have alluded to the population distribution of the country (age as well as rural/urban distribution) as one of the reasons for the relative success of Uganda viz a viz South Africa. Still the population density in Uganda is much higher.

Was there any data to support why the age/rural/urban mix was the reason why the disease spread more slowly.. for instance, were the cases more among the elderly or in the urban population

3. From the paper, we are not able to see whether there were any other factors such as implementational efficiency (state capacity), institutional arrangements, etc which resulted in Uganda doing a better job. You have described the governments in both states and also stated that the public supported the lockdown. Were there any features of state legitimacy in the months that followed that improved public compliance of social distancing? Were you able to elicit any of these from the qualitative interviews.

I feel these areas if elaborated with available data and interview response could strengthen the paper .

Reviewer #2: The investigators set out to compare physical distancing measures implemented in South Africa and Uganda during the COVID-19 pandemic in terms of their approaches and outcomes.

Here are some comments and recommendations for the authors to consider:

1. To properly situate the discussion on the impact of the policies, an exploration of differences in public health interventions that also impact the outcome needs to be considered. For example, what were the policies on use of face masks and access to testing. Whilst masking policies may have impacted infection risk, access to testing impacts case finding.

2. Potential differences in completeness of reporting needs to be accounted for if you are going to make inferences about the impact of public health interventions during COVID. Were there differences in timeliness and completeness of reporting between South Africa and Uganda?

3. There is some justification in the text, but I was not entirely sure of the rationale for comparing SA and Uganda, especially considering their income differences.

4. Overall, the manuscript is substantially longer than it needs to be. A lot of the material in the results can be displayed in tables instead.

Introduction

• Page 4 line 64-66: occupational exposure was an important factor particularly in the early days of the pandemic.

• Page 6 line 120-123: please provide references.

• Page 7 line 141-142: sentence beginning “contextual factors…” is redundant. Consider removing.

• Page 7 line 142-145: please clarify how you came to the conclusion on transparency in COVID-19 data collection and reporting.

Methods

• Please include the data collection form as an appendix or supplementary material.

• Please clarify if you sought (perhaps from the informants) unpublished memos or other documents that may have been relevant. Also applies to documents that may have been published in print or were otherwise unavailable on the internet.

• Please provide more details on your content analysis procedure. How did you manage codes or themes, for example? Details to enable the reproducibility of your results need to be provided.

• Please clarify if local ethics approval was obtained in SA and Uganda. Why was verbal informed consent used instead of written consent? What is the plan for sharing the findings with the participants?

• Did you conduct the key informant interviews in-person or remotely? If remote, what technology was used? What was the language of the interviews? How was the data handled? How was transcription conducted?

Results

• Did subnational levels of government have a say on the implementation strategy for the national mitigation/containment policies? Did that impact the outcome in some way? This will be good to know since you specifically excluded documents on subnational responses.

• Page 15 line 294: SA and Uganda are described as highly populated. What is this relative to?

• Page 16-17: line 330-340: I am not sure of the relevance of the discussion on global health security ranking given how far off the index was for predicting COVID-19 outcomes in the ranked countries. Consider deleting to cut the word count.

• It may be insightful for the reader if per capita health expenditure in SA and Uganda are provided.

• Page 18 line 365: what is the date for the Uganda inequality score?

• Page 19 line 387-388: linking HIV prevalence to immunocompromise may be incomplete without indicating the proportion of infected people who are on antiretroviral treatment.

Discussion

• There is an absence of a discussion on isolation policies for infected people. Were those not considered relevant for the epidemiology of COVID?

Reviewer #3: First of all i would like to declare no competing interests with the authors or their institutions.

This is a well written paper which is providing important information that can inform future pandemic responses on the continent and globally. Therefore the chosen research topic is relavant. However individual country responses are also complex which may require more extensive and comprehensive research.

South Africa and Uganda have different government systems, human rights profiles and freedoms. This informs responses to state led efforts. In contexts where there are authoriatrian elements enforcements and responses to restrictions may be limited due to suppression of freedoms. South African is also the 3rd largest economy on the continent and has some of the most advanced infrastructure and leading research institutions on the continent. This has implications on a comparative analysis. I would recommend the authors to include in the context of the two countries more information about health systems in both countries especially human resources , financing and public health infrastructure.

With regards to epidemiological data , our world in data is a very useful resource however data quality and comprehensiveness highly depends on publicly available data sources. The available data therefore may differ from country to country depending on individual country information systems and public platforms with available data. The authors did acknowledge limitations of relying on publicly available data but i would encourage them to triangulate data from our world in data with information from the regular situation analysis released by respective ministries of health.

The authors did well in comparing factors that contribute to possible severity and progression of the diseases which could potentially explain the mortality rates and severe cases recorded.

However when it comes to overal analysis of the impact of various policies on the progression of the pandemic its important to note that its complex and multifaceted influenced by various factors therefore a more in depth individual country case studies would have yielded more information that will be more useful for multi-country analysis

6. PLOS authors have the option to publish the peer review history of their article (what does this mean?). If published, this will include your full peer review and any attached files.

**Do you want your identity to be public for this peer review?** For information about this choice, including consent withdrawal, please see our Privacy Policy.

Reviewer #1: No

Reviewer #2: No

Reviewer #3: No

---

## [Decision Letter · Decision Letter 1]

3 Jun 2024

A comparative analysis of COVID-19 physical distancing policies in South Africa and Uganda

PGPH-D-23-02580R1

Dear Ms. Mohammad,

We are pleased to inform you that your manuscript 'A comparative analysis of COVID-19 physical distancing policies in South Africa and Uganda' has been provisionally accepted for publication in PLOS Global Public Health.

Best regards,

Veena Sriram

Academic Editor

Reviewer Comments (if any, and for reference):

Reviewer's Responses to Questions

**Comments to the Author**

1. If the authors have adequately addressed your comments raised in a previous round of review and you feel that this manuscript is now acceptable for publication, you may indicate that here to bypass the “Comments to the Author” section, enter your conflict of interest statement in the “Confidential to Editor” section, and submit your "Accept" recommendation.

Reviewer #1: All comments have been addressed

Reviewer #2: All comments have been addressed

2. Does this manuscript meet PLOS Global Public Health’s publication criteria? Is the manuscript technically sound, and do the data support the conclusions? The manuscript must describe methodologically and ethically rigorous research with conclusions that are appropriately drawn based on the data presented.

Reviewer #1: Yes

Reviewer #2: Yes

3. Has the statistical analysis been performed appropriately and rigorously?

Reviewer #1: N/A

Reviewer #2: N/A

4. Have the authors made all data underlying the findings in their manuscript fully available (please refer to the Data Availability Statement at the start of the manuscript PDF file)?

Reviewer #1: Yes

Reviewer #2: Yes

5. Is the manuscript presented in an intelligible fashion and written in standard English?

Reviewer #1: Yes

Reviewer #2: Yes

6. Review Comments to the Author

Reviewer #1: Thank you for the responses to my comments. I hope further documentation of the work done by countries all over the world, including in Africa and the rest of the developing world, will yield more insights into the factors that led to better outcomes during the CoViD 19 crisis.

Reviewer #2: (No Response)

7. PLOS authors have the option to publish the peer review history of their article (what does this mean?). If published, this will include your full peer review and any attached files.

**Do you want your identity to be public for this peer review?** For information about this choice, including consent withdrawal, please see our Privacy Policy.

Reviewer #1: No

Reviewer #2: No
